# The Driving Role of 3D Geovisualization in the Reanimation of Local Collective Memory and Historical Sources for the Reconstitution of Rural Landscapes

**Dimitris Goussios [1] and Ioannis Faraslis [2],***

[1] Department of Planning and Regional Development, University of Thessaly, Pedion Areos, 38334 Volos, Greece
[2] Department of Environmental Sciences, Campus Gaiopolis, University of Thessaly, 41110 Larissa, Greece
* Correspondence: faraslis@uth.gr; Tel.: +30-2410-684344

**Abstract:** The dynamics created by the process of territorial construction are partly based on the selective and functional incorporation of heritage. However, in rural areas, retrospection presents particular difficulties due to a lack of appropriate information. This research proposes the implementation of a methodology that combines sources, methods, and tools where the extraction of timeless information is based on the use of 3D interactive representations incorporating the active participation of actors and their collective memory. The proposed methodology strives for the compatibility, objectivity, and synergy of information from various sources and historical periods. The scope of this research concerns the mapping of the route and landscapes that were explored and described by the traveller Leake 210 years ago in the Farsala-Almyros area in Thessaly (Greece). The results focus on the reconstruction of the spatial subsystems of land use and exploitation at the beginning of the 19th century. The analysis reveals a production system, organized to use the laws of nature in order to sustainably manage the relationship between humans, animals, and natural resources. At the same time, the comparison with the current space has revealed a serious degradation in the natural environment since then. Finally, this mixed methodology, by combining the "spatialization" of information, virtuality and interactivity, the transition in time and space, and, finally, the "territorialization" of information, forms the basis for the inclusion of the history of places in the modern process of constructing a territorial area.

**Keywords:** 3D-geovisualisation; local collective memory; territorial construction; Ottoman period

## 1. Introduction

At a time when issues regarding nature, international competition, and quality are driving the readjustment of production and consumption patterns, more and more rural areas are turning to a territorial mode of development [1,2]. This new pattern marks the transition from raw material to resource [3] and the meeting of the economy with heritage, a driving force in the activation of territorial resources [4,5]. The competitiveness and value of these resources depends on their distinctiveness and this, in turn, on the dual correspondence between the material and immaterial nature of the resources and heritage of the place [6,7]. This interdependence explains the interest of rural areas in heritage [8–11]. In this context, retrospection contributes to highlighting the continuity of space as an evolving social product, enriching resources, and the identity of rural territorial areas [12].

This intentional search for functional elements of heritage, by future-orientated rural areas, broadens the scope of historical geography and cultural geography [13–15]. Historical retrospection is henceforth also linked to a places' own pursuit in reclaiming space and time in a process of social construction and control of their resources [10,16]. In an effort to re-approach the relationship between space and time, communities entrust heritage with strengthening the links between resources and their space [11,17,18]. However, since

heritage reflects the symbolic dimensions of the relationships between communities and their places [19], its integration in the territorialisation process cannot but be supported by the local community [20]. In this process, the role of heritage is expanded as a driver of identity and a mediator in the evolution of the relationship between nature and humans, thus acquiring a pedagogical role in the effort of local communities to create a sustainable model of development [21,22].

However, retrospection encounters certain difficulties in rural areas due to a lack of information and their morphological elements, in particular where the landscape has undergone radical changes [23]. The activation of local collective memory could help overcome these obstacles [24,25]. This is possible, if memory can act as a guide to connect various intermediate time periods, between the past and the present [26–28]. This finding is of importance as the virtual representation of the historical reference space and its incorporation with oral tradition are a prerequisite in the identification of the spatial dimension of heritage (e.g., locations where the merging of its intangible and material elements takes place) [29,30]. This effort can be supported by the use of a virtual interactive environment which can create further synergies between sources and memory by overcoming the limitations of an abstract 2D map [31].

These synergies can be supported by the continuous research featured in the international literature on the use of 3D geovisualisations as communication tools in the context of spatial planning [32,33]. Although most implementations using 3D models focus on the study of urban heritage and the historical urban landscape, their evolution has led to research on the activation of actors in the diagnosis of rural space [34]. For instance, the ability to switch between scales, the transitions in time and space of different time periods, and the interactive visualisation of space from different viewing angles are some of the advantages that 3D representations (maps, aerial photos, satellite images, and 3D models) offer in the diagnosis of spatial issues [35–40]. This progress in the accuracy and navigation of representations can, by enhancing interactivity, activate local collective memory in rural areas, though few implementations have used 3D geovisualisation for this purpose [41–43]. Furthermore, the creation of 3D geovisualisation of the rural paleo-landscape fosters the participation of actors in the reconstruction of past spatial organisation by facilitating the transition of information and the bidirectional movement of actors between two different time periods [44–47].

In this context, determined by territorial dynamics, this research organizes the implementation of a methodology which attempts to combine: (a) the immediate perception provided by 3D geovisualisation of the data from the description (in situ) and (b) the ability to alternate between visualisations using geospatial information (aerial photographs, satellite images, etc.) with different spatial and temporal resolutions (in visu) [48,49], and (c) the collective memory of local actors with a focus on the history of places [19,50]. By combining sources and tools, this methodology aims to create synergies that facilitate the transition of actors and information from the past to the present, and from virtual to real space. This process seeks to reveal "hidden components and/or resources" and leads to the "territorialisation" of historical information. The scope of this research is the mapping of the route and landscapes explored and described by the traveller Leake 210 years ago in the Farsala-Almyros[1] area in Thessaly, Greece.

## 2. Materials and Methods

### 2.1. Study Area

In the centre of the Greek territory, and specifically in the region of Thessaly, historical research has barely covered the Ottoman period, particularly in terms of the spatial organisation of small rural areas. However, new data, including easier access to the Ottoman archives and travel journals, are emerging to support research on this period. The English officer William Leake was one such traveller in Greece. On 10 December 1809, he travelled between the two small agricultural centres of Farsala and Almyros in the region of Thessaly (Figure 1) [51]. These centres control two small fertile plains which are dominated by pas-

toral livestock farming, cereal and fodder crops, and small irrigated areas with industrial and leguminous crops. The relief is smooth, with hilly areas. The area is crossed by the river Enipeas and a number of torrential streams.

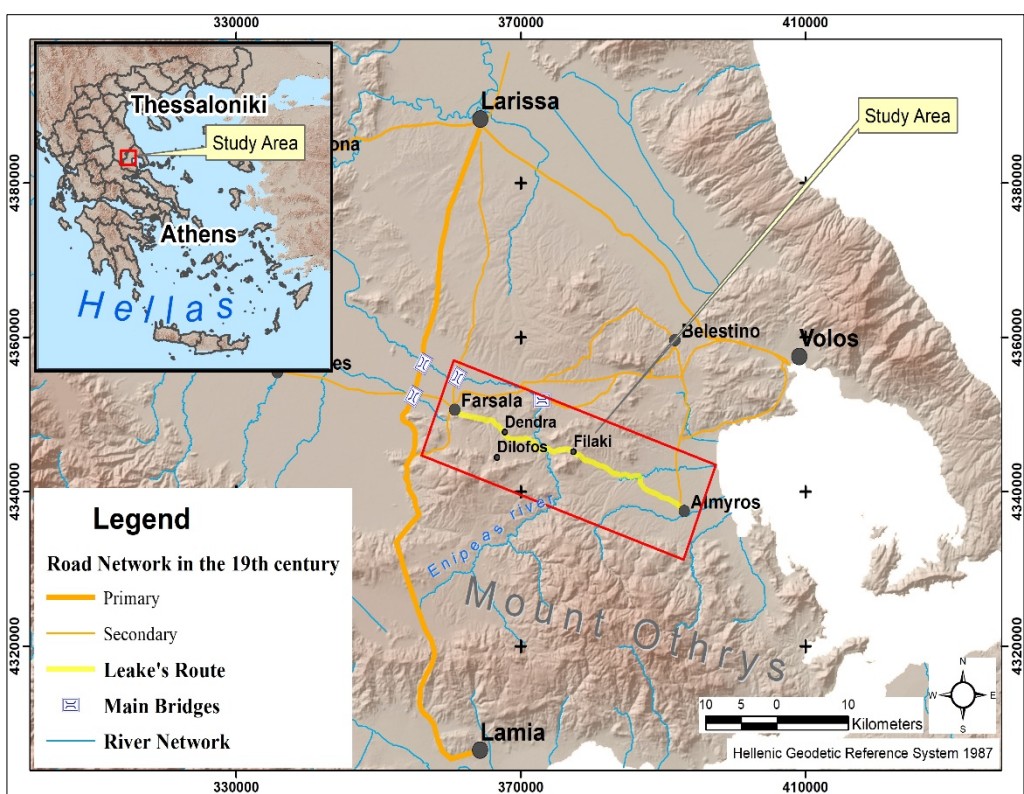

**Figure 1.** The area travelled by Leake in the 19th century.

The research was organised in two distinct phases. The first focused on mapping the route between the two rural centres, and the second on the visualisation analysis of land uses/covers and the landscape based on the Leake's observations.

*2.2. Materials*

Available cartographic and descriptive data for the study area were collected and processed, including:

- Topographic diagrams from 1970 (1/5000 scale) (ordered from the Hellenic Military Geographical Service, https://www.gys.gr/hmgs-geoindex_en.html, accessed on 20 November 2022). The contour lines (at four-meter intervals) were digitised creating the Digital Elevation Model. Additionally, toponyms and hydrographic and road networks were extracted in a vector layers format.
- Five aerial photographs (AP) from 1945 with a 1:42,000 scale (ordered from the Hellenic Military Geographical Service, https://www.gys.gr/hmgs-geoindex_en.html, accessed on 20 November 2022). This dataset are the oldest available aerial photos for the study area, representing the landscape almost as it was one century ago. The preprocessing and processing procedure, including the digitalisation and the creation of the orthomosaic map of the study area, was applied.
- Twenty aerial photographs from 2016 with a 1:5000 scale (ordered from the Hellenic Cadastre, https://www.ktimatologio.gr/el, accessed on 20 November 2022). Also, the final orthomosaic map was created.
- The agricultural census of 1911 [52]. It shows the farming systems and production before two radical reforms: (a) the expropriation of Chiftlik land and (b) the redistribution of agricultural land in 1925.

-       A field study to identify points on the route with local residents based on the traveller's narrative (landscape, activities, etc.).

*2.3. Methods*

The process of articulating and integrating the various types of information in space, through the reconstruction of spatial subsystems, is structured by combining three elements: (a) the identification of spatial structures (settlements, roads, farming system), (b) the creation and application of 3D interactive representations, and (c) procedures for the active participation of actors. The main components of the proposed methodology are presented in the flowchart below (Figure 2).

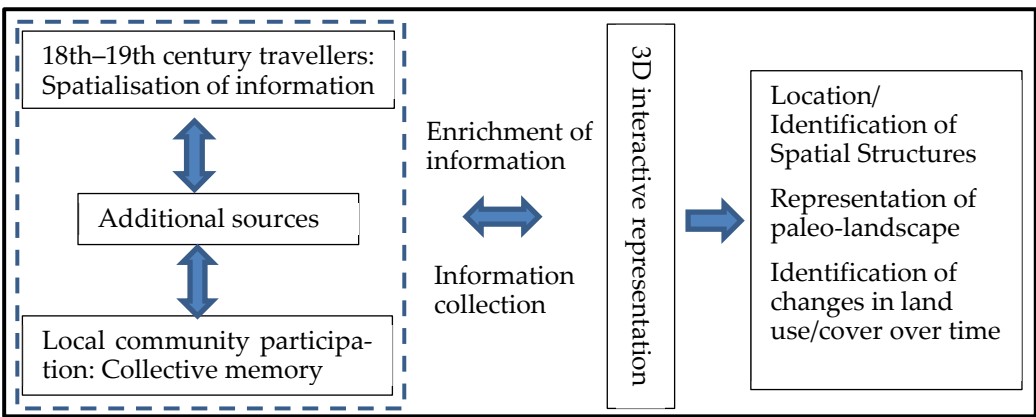

**Figure 2.** Flowchart of the proposed methodology.

In order to map the route and present the landscape as described by Leake, the methodology was organised in three phases:

➢      Identification of spatial structures and compatibility of information

The primary objective was to enhance the objectivity of early 19th century travellers' narratives and the use of the spatial dimension of their information through the mixed methodology. Firstly, Leake's reference space was determined, and the content of the narratives was analysed. The area explored was identified and the themes of the traveller's data were determined: land covers/uses, landscapes, and monuments, etc. Next, the positions, functions, and interdependencies between the material elements (houses, herds, castles, religious monuments, parcels of land, and bridges, etc.) and spatial structures were highlighted.

In the first phase, the proposed methodology attempts to incorporate all relevant information into the space, following a logic based on the structure and operation of spatial subsystems [53]. The choice of additional sources is guided by the initial attempt to integrate elements from the narrative. These sources enable the presentation of additional elements which, combined with the previous ones, reveal additional spatial links and interdependencies. The elements highlight spatio-functional entities (SFEs) such as: house-settlement, path-route, riparian zone-stream-irrigation-crops, and herd-pasture which are functionally integrated in the corresponding spatial subsystems (e.g., land-use, road network, etc.).

This integration process expands the ability to use information from other sources (travellers, statistics, A/Ps, local collective memory, and interviews, etc.). These diverse elements (sources and dates) raise the question of relevance and compatibility. Information is considered relevant when it refers to a position, form, or function within a spatial structure or a subsystem in the area. Moreover, this additional information is compatible with the traveller's information when it refers spatially to the same place and temporally to a period with the same land structures and technical means. In this case, information from the 1810–1911 period is considered compatible due to the fact that there has been no change in

land relations, and information from the 1810–1950 period as there has been no change in the infrastructure with regard to land structures and the production model. The combination of relevance and compatibility of the various sources of information enables the spatio-temporal and thematic combination of data. This produces additional information and knowledge about more complex concepts such as the farming and production system, the relations between land cover/use and natural resources, but also knowledge, practices, etc. Additionally, it contributes to the reconstruction of a spatial subsystem by emphasising interdependencies between elements from travel narratives (roads, land parcels, houses, and monuments) and other elements within it (e.g., agricultural census, local memory, etc.). The integration into the spatial subsystems of this 2nd level information (knowledge, know-how, practices, etc.) can lead to the identification of information of a territorial nature (heritage, site management, etc.).

➢　Creation and implementation of 3D interactive representation tools

The creation of 3D interactive cartographic data, at two points in time (1810 and 2020), constituted the basic model to reconstruct spatial structures (subsystem) and to compare the old landscape with the new. At the same time, reference points such as settlement positions and key toponyms in the area were used to enhance the model's functionality. All spatial information was coded and stored in a geographic information system (GIS). The spatial analyses of visibility, time distances, etc., were used as additional tools in the verification of information provided by diverse sources (e.g., travellers, local residents, etc.).

➢　Enhancing the synergy of information through the active participation of actors

The combination of methods and tools appears decisive in highlighting the systemic relationship between all available elements. The role of 3D representation is to contribute to the integration of information from the travel narrative into a network of complex interdependencies between geomorphology, physiography, and spatial structures (e.g., hydrographic network/irrigation system and crop orientation). The participation of local actors is key to the whole process. The 3D model is also a means of mobilising people who are considered bearers of information accumulated from older generations, through up to 200 years of oral tradition. The enrichment of the 3D model with information is achieved through the direct participation of actors from the first stages of its creation. The selection of local actors was based on the following criteria:

- Being old enough to remember the trails and the landscape of the 1950s and 1960s;
- Being a farmer who has lived mostly in the area, with good knowledge of information relating to the spatial structures;
- Being observant with good spatial perception.

Snowball sampling was adopted to select participants; in the first meeting residents were asked to indicate fellow residents with both knowledge of traditions and good orientation. Then, the final group of residents with the aforementioned traits was formed. In this case, the issue of reliability was not dependent on the number of observations (since there was not much dispersion), but rather on the association of the virtual landscape with the image in the collective memory. The 3D model was used to activate interactions between local residents and planners in a three-step participatory planning process (Figure 3). In Step 1, the local actors became familiar with the 3D visualization and were trained to recognize the elements of space. In this phase, the most appropriate stakeholders were selected according to the above-mentioned features. In Step 2, there was direct communication between stakeholders in the 3D environment. Current and historical information, especially regarding the spatial structures on Leake's route, was collected from them. The meetings included, inter alias, the projection of specific spatial units (e.g., the banks of a one-kilometre-long stream) with the participation of descendants of the original landowners. The use of ancient paths until the 1950s made their detection on the 1945 aerial photos possible. For parts of the path not visible on the cartographic data, information from residents was used. Next the information was coded, visualised, and presented in the 3D GIS interactive environment during meetings between the research team and local

residents. Finally, in Step 3, the 3D model was used for analysis (e.g., visibility analysis) and evaluation of the results. Changes to the road network, farming system, landscape, and natural environment were identified and substantiated.

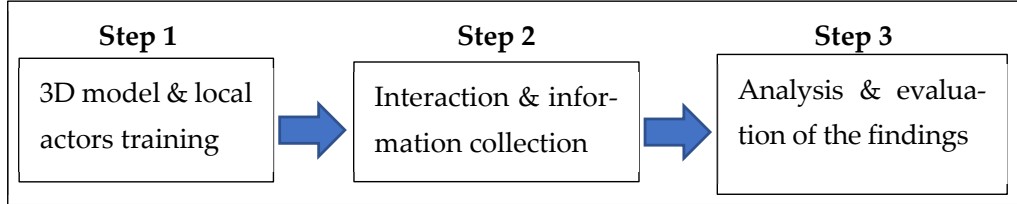

**Figure 3.** The workflow diagram of the 3D participatory process.

## 3. Results

### 3.1. Main Elements of the Route-Identification

Mapping of the Farsala-Almyros route was based on the identification of the positions and points mentioned by Leake. An almost straight route of 38.5 km emerged. The passable semi-mountainous zone was chosen to avoid inaccessible lowland areas due to flooding (Figure 4).

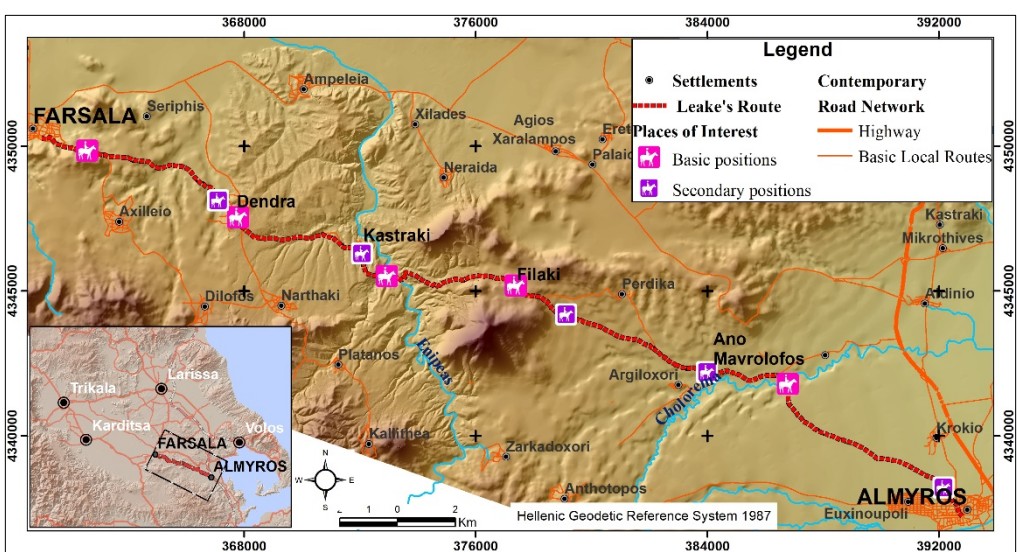

**Figure 4.** Mapping of the Farsala-Almyros route, as described by Leake.

Mapping shows that this was a path that did not allow larger groups of people and animals to travel in both directions simultaneously. The route from Farsala to Almyros followed successive paths which connected intervening settlements. These paths followed easy passages determined by the topography and hydrographic network and were carved along the borders between cultivated and heath lands. It was an interlocal road connection, for light loads, and of little importance. It is worth noting that the photographic interpretation of the A/Ps of 1945 identifies two different types of tracks on parts of the route (Figure 5):

(a) Cart roads: they had deep tracks and a greater width. They followed routes along the edges of hills and avoided steep slopes.

(b) Mule trails: they had faint tracks and were found at higher altitudes, following natural passes (small mountain passes, small rivers).

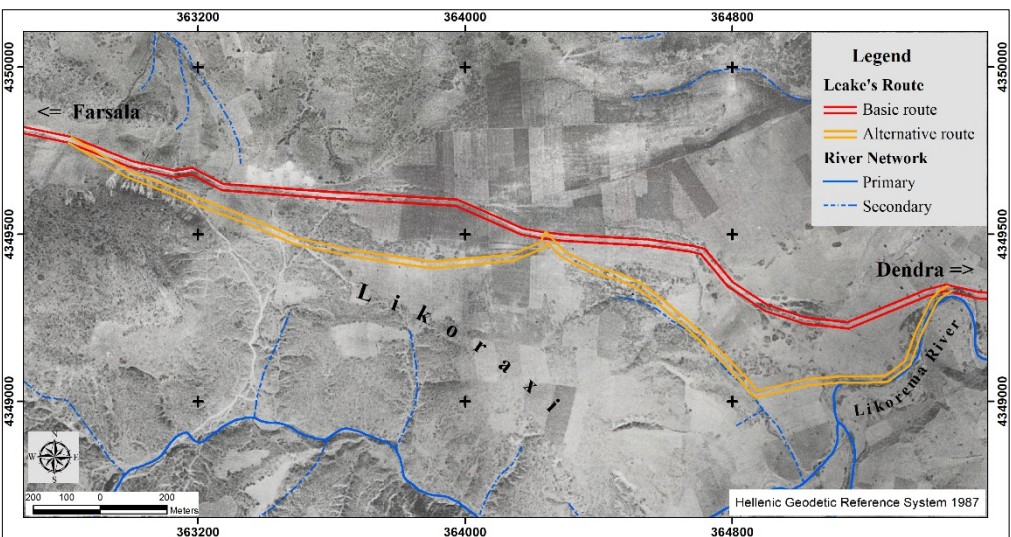

**Figure 5.** Section of the Farsala-Almyros route, as described by Leake.

This alternative route confirms the information that the road was used to transport heavy loads of wheat by cart from Farsala to the port of Almyros. Nevertheless, the absence of signs of maintenance and improvement by the central administration indicates that it was not a road "constructed" with the public interest in mind. This finding is reinforced by the fact that in difficult places, such as the Enipeas river and stream crossings, there is only an 18th century bridge in Cholorema, which is preserved to this day.

Analysis of the route mapping confirms research indicating that in the Ottoman Empire, roads deteriorated after 1600, impeding the development of trade between hinterlands and ports. Carriage roads were almost non-existent and nearly all land transport was carried out by pack animals on paths. In the case of the Farsala-Almyros route of the Leake period, the absence of interventions on the road surface of the entire route, and the lack of bridges crossing the river Enipeas, reveals that there was minimal trade between the two centres: the hinterland and the small port. In 1809, this route remained part of a small-scale transport network.

### 3.2. Locating and Identifying Positions on Either Side of the Route

Identification of the route formed the basis for locating and verifying specific toponyms, positions, and monuments mentioned in Leake's description. The research investigated: (a) the Enipeas river crossing, (b) a monument that no longer exists, and (c) locations of special functionality (streams-irrigation, grazing-herds, and landscapes, etc.) in the organisation of the production system.

➢ Enipeas river crossing

In order to locate the point where Leake crossed the river Enipeas, his travel diary mentions that after the village of Kastraki he followed a path, which after crossing the river, ascends to the next village of Filaki, following a route between two hills. This information is confirmed by the toponym "Passage" which exists on modern topographical diagrams, as well as by field research through 3D modelling (Figure 6).

➢ The "Dervish School" monument

Today, the absence of ruins makes the monument difficult to locate. The search for the location of a Muslim "Tekke" ("Dervish School") was based on Leake's account of the position, orientation, and distance from the observation point. The reference to gardens indicates the existence of a source. To identify its position, a simulation (visibility analysis technique) was used which facilitated the participation of residents (Figure 7).

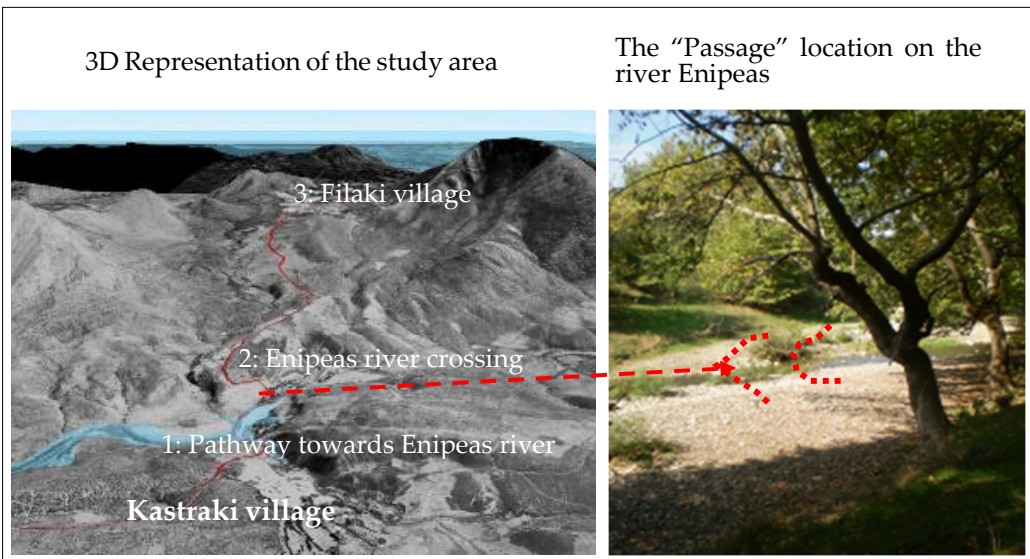

**Figure 6.** Validation of the "Passage" location on the river Enipeas.

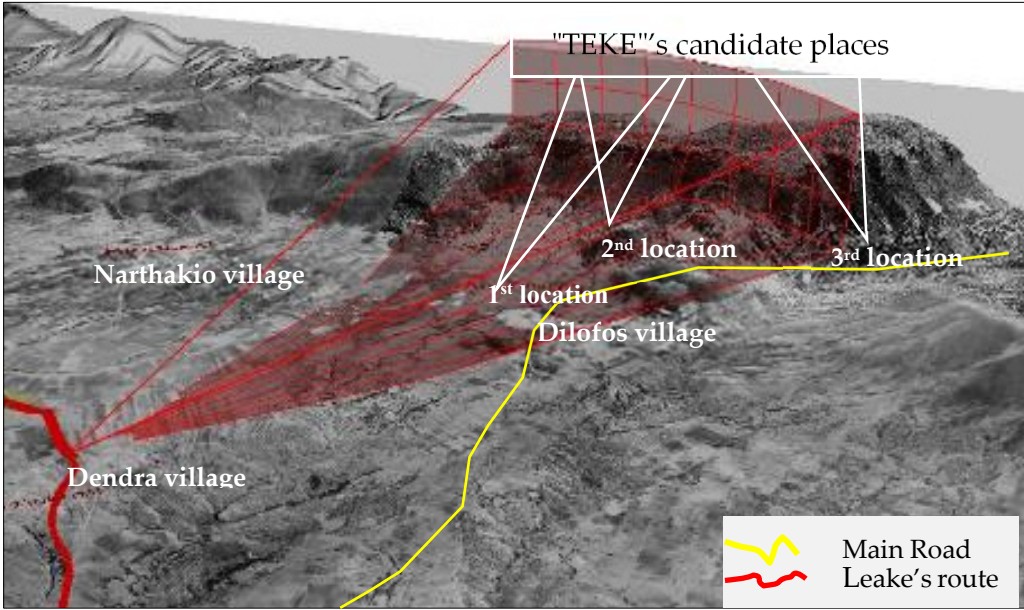

**Figure 7.** Visibility analysis and the 3 possible locations of the "Tekke".

An abundance of related (baths, vakuf), but unknown, information emerged, based on which three possible locations for the "Tekke" monument around the village of Dilofos were examined:

- 1st & 2nd location: residents' accounts of the existence of Ottoman baths and the toponym "Tekkes" (the residents call it the vakuf of "Tekke") on the outskirts of the village of Dilofos indicate the existence of a tekke, but not its position.
- 3rd location: the "Tekke" is located three km southwest of Dilofos on the road to Domokos.

The 3rd location ultimately seems the most likely as: (a) Leake located the "Tekke" two miles further away from the settlement, whose position has not changed, and (b) "Tekke" monasteries were built on important road axes and not in or next to villages.

### 3.3. Identifying Structures and Functions of the Paleo-Landscape

Leake's information, combined with other sources (censuses, A/Ps, travellers, and research), interpret elements of the 19th century landscape on either side of the route, allowing a first spatial logic to be projected on a micro-scale (Table 1). More specifically, the knowledge that the journey took place at the beginning of December contributed to both the interpretation of the landscape, as well as the identification of the crops and farming system. At this time of year, specific colours are reflected in the land cover and in the parcels of the farming system. Ploughed fields have just been sown with grain or have been prepared for spring crops (in black) and can be distinguished from the uncultivated fields integrated in a crop rotation system. The following table shows how the combination of this information allows its integration in spatial structures, highlighting the inscription of the farming and production systems on the landscape and space (Table 1).

**Table 1.** Categorisation of 19th-century landscape elements.

| Sources | Farming System | Agricultural Production | Livestock | Landscape |
|---|---|---|---|---|
| Travel narrative | Distribution of cultivated and unploughed land | Crops on the riverbank | Herds and types of animals, grazing | Vegetation, hydrographic network, land cover, settlements |
| Statistics 1911, 1881 | Fallow land, agro-sylvopastoral | Leguminous, food and fodder crops | Types and numbers of animals, production | Small increase in cultivated land |
| 1945 Aerial photograph | Distribution of parcels, Crop rotation | Parcels of land, size, in valleys, orientation | Pastures | Expansion of farming due to reform |
| Other sources: Sivignon M. 1975, [54] | Organisation of crop rotation per holding | Irrigation from the river | Herds, quality, value | Land clearing, levelling (satellite images) 2000–2020 |
| Collective memory | Crop rotation per holding, fallow land | Orientation of crops/grazing, irrigation method | Grazing system | Land clearing, levelling |

Leake's reference to the existence of ploughed land on the banks of streams provides both land use and position. The combination of the A/P of 1945 and collective memory provides additional information regarding the form, sizes, and ownership of the parcels. The alternation between ploughed and unploughed parcels reflects a form of crop rotation in the riparian zone [54][2] and confirms later sources (Figure 8).

This zone featured a cultivation and irrigation system integrating crop rotation. On a community level, the crop rotation system was organised at a ratio of 30–40% of fallow land in relation to cultivated land. It is estimated that in 1809 there was less arable land than in 1945, while much of the area was devoted to grazing.

Irrigation was based on the hydrographic network, avoiding steep slopes (water retention). The road represents the boundary between cultivated land and the stream. There was a distance between the two to avoid flooding, which created a biodiversity and grazing zone. This zone was defined by the repeated allocation and layout of parcels of land (cover—ploughed or fallow, slope, and orientation) as elements of the farming system and the structure of the landscape. It is noted that the change from winter to summer informed the choice of northeast (food crops) and southeast (pasture, uncultivated land) orientations for early morning grazing in winter. In general, the dependence of agricultural activity on the geomorphological and physiographic characteristics of the area is evident. At the same time, the mosaic of alternations between crops—fallow land, pastures, and heaths—described by Leake is verified (Figure 9).

All these sources confirm the abundance of natural resources (especially water) and emphasise the close relationship between agriculture and pastoral livestock farming within the land regime (Chiftlik) and the mixed extensive production system.

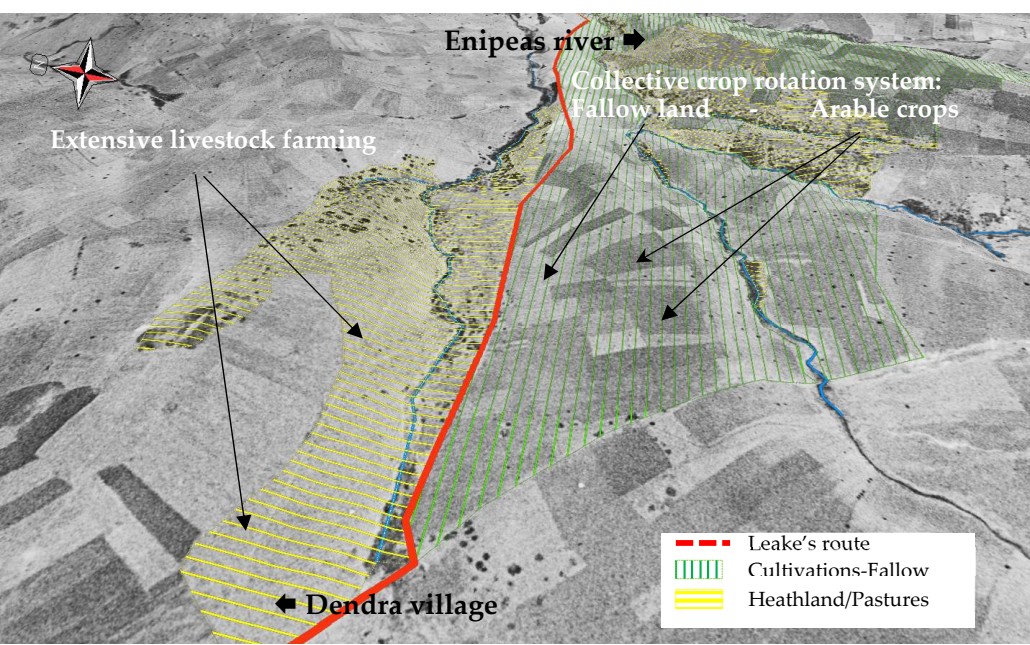

**Figure 8.** Depiction of the mixed agro-livestock production system.

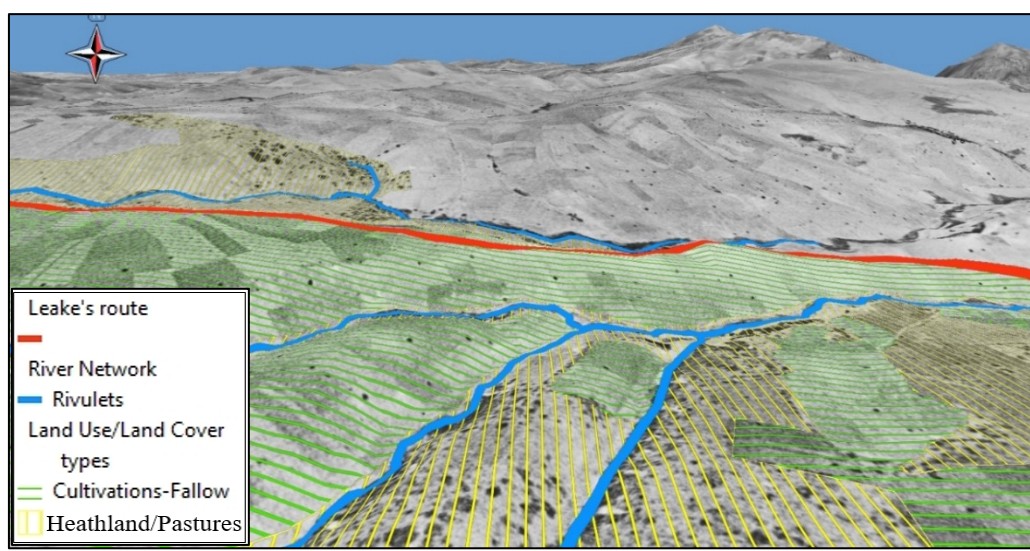

**Figure 9.** Depiction of the land cover mosaic (crops, pasture, and natural vegetation).

### 3.4. The Landscape and Spatial Systems in the Early 19th Century and Their Evolution

The synergies of the methodology generated the conditions to produce a wealth of information of a historical and ethnographic nature, integrated in the space and its structures. The participation of local actors in all steps of the methodological chain helped both: (a) to fill the gaps concerning the spatial and land structures, where this was not possible through other sources, and (b) to evaluate and validate the landscape changes that occurred in the period under consideration. A consensus between local residents and the research team was reached during the meetings. This consensus was based on the accurate geospatial mapping of the past and present landscape, which helped residents understand the catastrophic changes that have occurred, linking them to impacts that are visible today (dried up streams, floods, soil erosion, and loss of biodiversity).

### 3.4.1. The Landscape and Spatial Systems at the Beginning of the 19th Century

Furthermore, the accurate mapping of Leake's route enabled the localization (geolocation) and interpretation of his descriptions of the surrounding landscape. These synergies provided details on a small scale, regarding the operation of an interlocal route, with difficult sections adapted to transport heavier loads to local ports. The information contributes to research on the role of an early 19th century interlocal road connection between the hinterland and the ports of Thessaly, as well as to the recognition of the route as a cultural heritage resource.

The analysis of the landscape based on the route provided an overview of spatial organisation at the community level, in the context of the Ottoman Chiftlik land regime and family farming. A residential network emerged of villages with an average population of 200 inhabitants, 4–5 km apart, with houses and roads in poor condition. The activation of local collective memory in a 3D interactive environment contributed to the interpretation and enrichment of historic non-spatial information, such as in the case of the "Tekke" and the farming systems. Despite the lack of traces/ruins, the process located and identified components (Ottoman baths, vakuf land, etc.) of a hidden resource connected to the religious "Tekke" monument and the presence of a Muslim population in the area until the early 20th century.

The identification of the position and function of various elements in the space enabled the virtual representation of the spatial subsystems of land use and cultivation. The management and cultivation of pastures and agricultural land was organised on the basis of a mixed production system, integrating farmers, pastoral livestock farmers, and nomadic pastoralists [54]. The synergy of the sources provided a picture of the 19th century landscape, the distribution of land uses, and the organisation of production systems in Ottoman Thessaly. The methods of organising agricultural production, pastoral livestock breeding, and fallowing were represented in the space. The best land, based on orientation, gentleness of slopes, soil quality, and proximity to sources for irrigation, was suited to agro-foods. The rest of the land was dedicated to pastoral livestock farming. It combined the production of products such as corn, peas, etc., for animal feed, the use of permanent pastures, and an abundance of fallow land. The central role of livestock farming in the natural fertilisation of agricultural land and its contribution in food towards a family's survival (dairy, meat, wool, and leather) emerges. The mixed production system ensured high soil cover and biodiversity enhancement. It was a production system that was intelligently organised to use the laws of nature in order to sustainably manage the relationship between humans, animals, and natural resources.

### 3.4.2. Evolution of Landscapes over Time

Next, changes in the landscape were investigated in terms of plant cover and available agricultural land. From this inquiry, it appears that the land redistribution of 1925 was followed by land clearing, which intensified after 1965 [52]. The following figure shows changes over time, in the village of Dendra, resulting from the intensification of agriculture (Figure 10). Local residents evaluated these changes by comparing the two spatial representations, i.e., that of the Leake period and the current space, in a virtual interactive environment. This comparison led to an understanding of the changes in the production system and the landscape of the region. It also emphasised the impact of major interventions (expansion of arable land, reforestation, etc.) in environmentally vulnerable zones (e.g., steep slopes, limited natural cover). The figure clearly shows a large reduction in natural vegetation, soil erosion on steep slopes, particularly near stream beds, while crop rotation has almost been abandoned. At the same time, the need for irrigation increased, which led to the search for water through boreholes of ever-increasing depths.

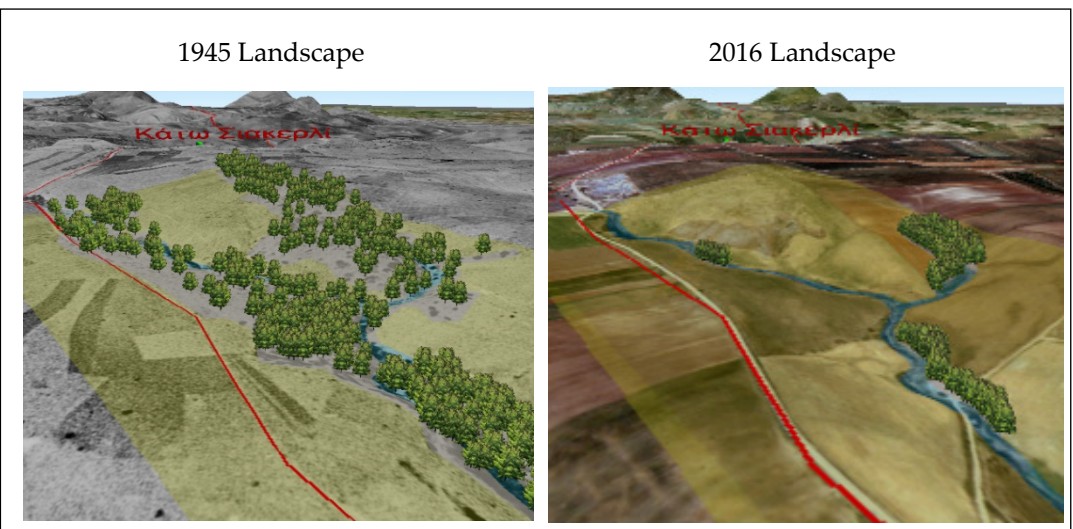

**Figure 10.** 3D Representation of the changes in the landscape in 1945 and 2016, respectively.

The above comparison allowed residents to highlight the loss of a rich biodiversity in the area, especially in riparian zones, and the decrease in water resources by emphasising the reduction, or even cessation, of the flow of streams for a large part of the year.

The creation and visualisation of relevant information also helped residents approach the new agro-ecological measures (eco-schemes) of the Common Agricultural Policy (CAP) as tools to mitigate impacts on the natural environment, soil, and water in the whole region, rather than only within the boundaries of their holdings.

## 4. Discussion and Conclusions

### 4.1. Discussion

The proposed methodology aimed to respond to the special difficulties encountered by any retrospection in future-orientated rural areas which seek to strengthen their territoriality. These difficulties are linked to the subjectivity of travel narratives, a lack of information, and the morphological characteristics which, in rural areas, rarely remain unchanged over time (landscape, land cover, and spatial regulations, etc.). This paper attempted to address these obstacles and the gaps in the literature by noting that while the reconstruction of historical landscapes is well documented in many research papers, especially from a technical point of view (3D and 4D digital models, web-mapping, and photogrammetric modelling), few papers refer to participatory processes with the simultaneous activation of local collective memory through the use of 3D communication tools [55–57].

This research implemented a combination of three types of information (narrative, related sources, and local collective memory) and 3D modeling through a methodological organization of steps as illustrated in Figure 11. It presents a network of successive interactions and synergies, as well as the transition from the "perceived" space of collective memory to the "real" reference space of local actors (Figure 11). These synergies counteract the ambiguities and incompatibilities between the various sources of information, which are thus transformed into systemic meta-information linking position and function, space, and time. This entire methodological approach creates suitable conditions for the identification of links between heritage and territorial area [9]. These links reveal the contribution of intangible heritage in the process of constructing territorial resources while, simultaneously, enhancing the ability of communities to control their relationships with resources and therefore territoriality itself [3].

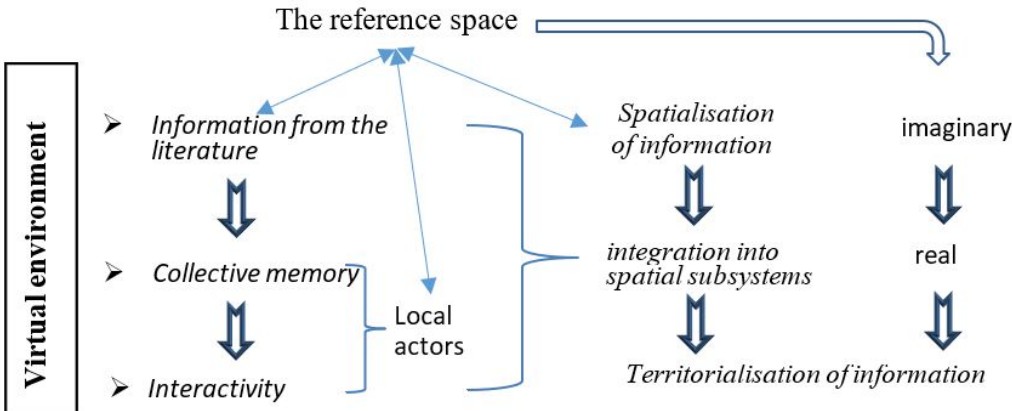

**Figure 11.** From the 3D virtual environment to the territorialisation of information.

This mixed, interactive methodology incorporating a feedback loop contributes to this process of transition in space and time through the following thematic areas:

➢    The spatialisation of information

For the spatial retrospection, information from three dimensions was combined: spatial (position, area, morphology, and orientation), functional (organisation of crops, irrigation, etc.), and temporal (alternating cultivation, crop rotation, livestock farming, and fallowing). The input of this data was made through the search for spatial structures in the landscape. Initially, the spatial dimension of travel narratives was used as a guide to integrate additional information based on thematic relevance and the logic of interdependence between the elements of the spatial subsystem of land use/cultivation. Next, the methodology focused on reconstructing the farming system. The resulting spatio-functional entities and systemic relationships (roads, settlements, irrigation perimeters, and fallow land, etc.) contributed to the description, and then to the virtual representation of a mixed production system of pastoral livestock farming and extensive agriculture, also emphasising its relationship with the natural environment.

➢    Virtuality (virtual environment), collective memory and interactivity

During the process of reconstructing (3D modeling) the spatial structures, systems, and landscapes, information of the various sources (traveler narratives, etc.) was utilized, contributing to the gradual, faithful representation of the space. This fostered the participation of local actors and the activation of their collective memory. This process was effective due to its high descriptivess[3] in terms of reality, representations, and the transformation of a static virtual space into an interactive virtual environment. The 3D GIS (using recent and past A/Ps) model enabled the bidirectional movement of actors between present and past space and time. This integration in the virtual space was supported by the combination of 3D representation of the landscape and the various spatial structures (in situ) with mediation visualisations (in visu). By operating between actors and the image, 3D models ensured interactivity, allowing actors to familiarise themselves with the use of the image, rather than being mere spectators. In this environment, the ability to switch between or mix scales in space and time helped actors recognise elements of the landscape through reference points and knowledge of their space. The activation of their memory completed the functional elements of the space provided by the travel narrative and other sources and showed where and how these connect to other positions and information. It also contributed to the virtual representation by participating in the identification of spatial structures and interpretation of additional information. Three dimensional representations and virtual navigation reconcile the two sides of space, the physical and the intellectual, and provide collective memory with a historical reference space, which no longer exists, thus allowing this memory to function as a source of more objective information.

➤　The reference space: from perceived to real space

The synergy of information and local memory in a 3D virtual environment enables actors to recognise the virtual paleo-landscape as a surface organised by their ancestors, and to then begin the process of reconstituting spatial subsystems of land use and cultivation. It represents a process of transition from the perceived space of collective memory to the real space. This process is based on the ability to interpret spatial organisation and management more objectively. This is due to the fact that the following are tested and integrated into the space: (a) the various synecdoches attributed to the signified of the words and phrases of the narratives; for instance, herds signify the pastoral system and the unploughed agricultural land signifies a crop rotation system and not to a permanent pasture, etc., and (b) images and practices from the collective memory, in combination with information from other sources (how crop rotation, irrigation, and grazing systems, etc. are organized). The generated meta-information provides additional knowledge about objects (often without traces) and practices (interdependencies, crop rotation, hydrographic network-irrigation, and grazing, etc.) and completes the puzzle of positions, forms, and functions of elements in the spatial subsystems of land use and cultivation in the 19th century.

This connection of forms, functions, and spatial structures highlighted the actual landscape as a part of the space which is recognised by actors and, also, exists independently of Leake's account. This landscape and the forms of spatial organisation inscribed on it form two sides of the same reality [23].

➤　The "territorialisation" of information

The territorial dimension of information is determined by its contribution to the process of constructing territorial resources and the territorial area itself. This process also incorporates "heritagization". In this instance, information (knowledge, practices, and values, etc.) is mentioned which highlights the physiognomy and identity of the area by combining natural heritage and human actions [58].

The role of 3D models in the process of "territorialisation" of information has proven to be essential. It is linked to the support of participation: (a) of actors in the bidirectional virtual space-time path between the present and the past, and (b) of local collective memory in the identification of current or potential links between the territorial area and heritage. A process emerges which, by uniting two time periods in one space and using time as the fourth dimension, expands and activates the spatial dimension of collective memory and transforms it into a source supplying territorial organisation with elements of timeless value (knowledge, management practices, traveller routes, and conservation of natural resources).

Finally, it seems that this mixed methodology, by articulating the above thematic fields, manages to propose a way to integrate the history of places in the modern process of construction of a territorial area. This methodology, by including the local community with the help of the virtual environment, ultimately produces a territorial information system, since collective memory intervenes in its two main components: (a) the geometric (position and form of elements), and (b) the semantic (function, intelligence of the landscape, and continuity with the present, etc.). Local collective memory, therefore, has a central role in the production of "territorial" information as a third source of information after that of the travel narrative and other sources. The actors, being a part of this memory, move between the historical and present space and become guides in the identification of links between territorial resources, heritage, and identity.

Notwithstanding the advantages resulting from the proposed methodology, some limitations emerged. There is a need for a multidisciplinary research team in order to support the three phases of the methodology. Three dimensional modeling, aerial photos, and public participation experts need to collaborate effectively with each other and with the local residents. Also, another limitation arises when there is a lack of old aerial photographs. Nevertheless, other historical maps could be used in the 3D model. From a technical point of view, ongoing research should focus on the development of more flexible and easier to

use 3D GIS models based on open source GIS software. This will allow a cost reduction in time and money.

In any case, the application of this mixed methodology creates synergies between the multiple sources of information, contributing to the highly accurate localization of spatial elements, structures, and systems. Additionally, this could help those who would like to undertake a reconstruction project of historical landscapes.

### 4.2. Conclusions

The implementation of the mixed methodology and the presentation of its results confirm, on the one hand, its effectiveness in the search for elements from the past that can be used in the future. Through innovation, it actively incoroporates the local community and its collective memory by combining virtuality, synergies between information, and bidirectional movement between the present/past and virtual/real space, in order to highlight links between heritage and territorial resources. On the other hand, it confirms the essential contribution of this methodology to the efforts of rural areas in redefining the relationship with their history based on their repositioning in the world. These areas, by building on their heritage-based specificity, can gain the ability to interpret the past through this methodology, which can then contribute to the process of territorial construction.

The methodological results indeed show that they offer a territorial structure support in the territorial approach-diagnosis of its historical places. This is possible because a territorial information system is produced which allows the region to capitalise on its experiences and preserve its knowledge, thus enriching territorial intelligence. Additionally, both the timeless values and inherited knowledge associated with sustainable practices (e.g., agro-environmental management systems), as well as the ability to compare virtual historical landscapes with current degraded landscapes, give this information a pedagogical value. This can contribute to the acquisition of common knowledge and the understanding of the relationship between past and present in the territorial area, and of the local applicability of sustainable development policies, or even to strengthening the resilience of agriculture in the face of climate change [59]. This territorial dimension of information is perceptible due to the contribution of representations and, by transferring experiences and knowledge, is also compatible with the need of rural territorial areas to capitalize on inherited experiences and promote the "territorial anchoring" of their resources [22].

Ultimately, this set of information produced by the mixed methodology, as elements of the specificity and/or guarantee-anchoring of territorial resources, can feed the fields of intervention of territorial engineering. Ensuring a combination of "endogenous" information production, collective intelligence, and territorial engineering can support the path proposed by the methodology to integrate the history of places into the modern territorial construction process.

**Author Contributions:** D.G. and I.F. conceptualized the research, performed the validation, and wrote the manuscript. All authors have read and agreed to the published version of the manuscript.

**Funding:** This research received no external funding.

**Conflicts of Interest:** The authors declare no conflict of interest.

### Notes

[1] They belong to the category of nahiyah (towns) and are the seats of kazas (basic administrative division of the Ottoman Empire). During that period, Almyros had a population of 2000 and Farsala, 2500.

[2] In semi-mountainous areas, crop rotation is organised in rows or in the centre of the parcel of land.

[3] Ability to describe, illustrate and present in a direct, vivid and expressive way.

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
