# Peer review of "The Driving Role of 3D Geovisualization in the Reanimation of Local Collective Memory and Historical Sources for the Reconstitution of Rural Landscapes"

_land, doi:10.3390/land12020364_

Round 1

Reviewer 1 Report

The authors present a well-structured exercise with a solid foundation, although, in my opinion, some variables, in addition to those presented, have to be taken into account.

It is certain that the exploration of the area under study, in the 19th century, could and would be more rational and more environmentally friendly. But it is also true that human resources are not the same, and that the logic of farming has changed.

Furthermore, climate change and its consequences are not being taken into account. Having those in mind, I'm not sure if many of the solutions from 200 years ago would be the most suitable today; only the authors are in possession of data to contradict, or agree with, this challenge/problematic.

Reviewer 2 Report

The paper explains in great detail the implementation of a methodological framework in which territorial diagnosis is based on the use of 3D-geovisualtions as communication tools in the context of spatial planning. 

The content of the article can be interesting even for a non-scientific reader indeed, and there is no lack of detail in the description of the whole process.

Finally, figures (maps) 1, 2, 3 and 4 should have a grid, giving the coordinates of the mapped area.

Reviewer 3 Report

The concept of linking a travelogue from a traveller 210 years ago to present day landscapes and its evolution over time is an interesting concept and can be a valuable part of the identity of the landscape, however, I have major concerns over the way the manuscript is constructed. 

The abstract needs to briefly cover, introduction, methods, results, discussion and conclusion to highlight to the readers the main contents of the paper, currently it reads mainly as an introduction with only vague references to the methods, results, discussion and conclusion

The introduction is vague and lacks a structured storyline.

Methods section:

What are signifiers and signifieds?

What is the difference between a material element and a spatial structure to me they are the same thing. 

It is not clear how the local actors are involved, what they were doing or what memory was being activated. Current inhabitants cannot remember an event from 210 years ago, so what were their memories connected to? What are the demographics of the local actors? How were they recruited? Did the local actors identify the changes or was this done by the researchers?

A workflow diagram showing the procession of work would aid understanding.

Results

Local residents evaluated the changes, but what was the reult of their evaluation? It seems to be only vaguely defined, whereas an evaluation suggests a degree of detail.

Discussion:

What is meant by "to strengthen their territoriality"? This could be understood differently depending on a researchers background and therefore needs defining.

I feel that the phrase "imaginary space" is being misused, as it suggests a made up space, such as a storyteller or designer would construct.

How did the information from the literature impact the collective memory?

Synedoches - "figures of speech"?

What does this mean? "A process emerges which highlights the “activatable” heritage (practices, knowledge, etc.), unites two times in one space, expands spatial memory and knowledge and enables the evaluation of the past through the needs of the future."

Conclusion

Place consumers is a term that suddenly appears in the conclusion. What is the relevance to the text and why is there a need for mediation between them and local residents?

I think you mean "knowledge" not "intellgence".

"It meets the development of a top-down territoriality that refers to values of nature once respected in the region, which are recognised within elements of its territorial identity with the help of collective memory" - How?

"The past is called on to contribute to the design of a sustainable agro-environmental model in the post-intensive period of exploitation of natural resources." Sounds nice, but how does this work in practice?

Reference 55 & 56, should be in the introduction and not first mentioned in a conclusion.

"Its work is facilitated through the proposed methodology by creating bridges with the past in order to reinforce territorial intelligence knowledge" Good

"with elements of timeless pedagogical value inscribed on the landscape and collective memory, and to incorporate them in the education of young people and the participatory planning of sustainable development in general." Not well explained what is meant by the timeless pedagogical vale or why it is necessary for the education of young people.
